# Excitatory and inhibitory D-serine binding to the NMDA receptor

**Remy A Yovanno[1], Tsung Han Chou[2], Sarah J Brantley[3], Hiro Furukawa[2], Albert Y Lau[1]\***

[1]Department of Biophysics and Biophysical Chemistry, Johns Hopkins University School of Medicine, Baltimore, United States; [2]W.M. Keck Structural Biology Laboratory, Cold Spring Harbor Laboratory, Cold Spring Harbor, United States; [3]Department of Biology, Johns Hopkins University, Baltimore, United States

**Abstract** N-methyl-D-aspartate receptors (NMDARs) uniquely require binding of two different neurotransmitter agonists for synaptic transmission. D-serine and glycine bind to one subunit, GluN1, while glutamate binds to the other, GluN2. These agonists bind to the receptor's bi-lobed ligand-binding domains (LBDs), which close around the agonist during receptor activation. To better understand the unexplored mechanisms by which D-serine contributes to receptor activation, we performed multi-microsecond molecular dynamics simulations of the GluN1/GluN2A LBD dimer with free D-serine and glutamate agonists. Surprisingly, we observed D-serine binding to both GluN1 and GluN2A LBDs, suggesting that D-serine competes with glutamate for binding to GluN2A. This mechanism is confirmed by our electrophysiology experiments, which show that D-serine is indeed inhibitory at high concentrations. Although free energy calculations indicate that D-serine stabilizes the closed GluN2A LBD, its inhibitory behavior suggests that it either does not remain bound long enough or does not generate sufficient force for ion channel gating. We developed a workflow using pathway similarity analysis to identify groups of residues working together to promote binding. These conformation-dependent pathways were not significantly impacted by the presence of N-linked glycans, which act primarily by interacting with the LBD bottom lobe to stabilize the closed LBD.

**\*For correspondence:**
alau@jhmi.edu

**Competing interest:** The authors declare that no competing interests exist.

## Editor's evaluation

Activation of NMDA receptors requires two co-agonists: Glutamate that binds to the GluN2 subunit and glycine/D-serine that binds to the GluN1 subunit. In the present manuscript, the authors address the interaction of D-serine, which is a less studied co-agonist than glycine, with the GluN1 and GluN2A subunits using molecular simulations as well as electrophysiology experiments. Surprisingly they find that D-serine interacts with the GluN2 subunit, further expanding our molecular understanding of NMDA receptor structure-function. This paper will be of interest to those who study NMDA receptors and ligand-gated ion channels in general.

## Introduction

The N-methyl-D-aspartate receptor (NMDAR) is an ionotropic glutamate receptor (iGluR) that uniquely requires the binding of a co-agonist in addition to its primary agonist for activation (*Hansen et al., 2021*). This heterotetrameric ion channel comprises at least two different subunits, GluN1 (isoforms 1–4 a and 1-4b) and GluN2 (subtypes A-D), assembled as a dimer of GluN1/GluN2 heterodimers (*Karakas and Furukawa, 2014*; *Lee et al., 2014*). The GluN2 subunit binds the neurotransmitter glutamate, while the GluN1 subunit can either bind the co-agonists glycine or D-serine.

Traditionally, glycine had been considered the major GluN1 agonist (*Johnson and Ascher, 1987*; *Forsythe et al., 1988*; *Kleckner and Dingledine, 1988*), but more recent work has suggested that D-serine may in fact be the dominant co-agonist for synaptic NMDARs in the brain (*Papouin et al., 2012*). D-serine is synthesized by the enzyme serine racemase expressed in astroglia (*Wolosker et al., 1999*) and neurons (*Miya et al., 2008*; *Balu et al., 2014*) and is released into the postsynapse by the Asc-1 transporter (*Rutter et al., 2007*; *Coyle et al., 2020*). D-serine binding to these synaptic NMDARs is responsible for inducing long-term potentiation (LTP), which is critical for memory functions (*Henneberger et al., 2010*). In addition, recent clinical efforts have indicated that D-serine could be a promising therapeutic for the treatment of neuropsychiatric disorders (*Peyrovian et al., 2019*; *MacKay et al., 2019*), most notably schizophrenia (*Kantrowitz et al., 2010*) and post-traumatic stress disorder (PTSD) (*Heresco-Levy et al., 2005*). Unlike the more well-studied agonists glutamate and glycine, the role of D-serine is less defined, causing it to be known as the 'shape-shifting' agonist (*Coyle et al., 2020*) that can adopt different roles in neurotransmission.

Each NMDAR subunit consists of an amino-terminal domain (ATD), a ligand-binding domain (LBD; also called an agonist-binding domain, ABD), a transmembrane domain (TMD), and a disordered cytoplasmic C-terminal domain (*Mayer, 2017*). The LBDs adopt a bi-lobed clamshell architecture that closes upon agonist binding (*Yao et al., 2013*; *Jespersen et al., 2014*). The conformational transitions of the LBDs from open to closed clamshells result in the generation of tension in the LBD-TMD linkers, which in turn facilitates gating of the TMD channel (*Tajima et al., 2016*; *Chou et al., 2020*). The ATDs allosterically regulate channel activities in a subtype-dependent manner via distinct interactions with the LBDs (*Yuan et al., 2009*; *Gielen et al., 2009*; *Karakas et al., 2011*; *Tajima et al., 2022*). Therefore, the LBDs can be considered the fundamental vehicles for driving ligand gating. Previous computational studies of NMDAR LBDs have indicated that glycine binding to the GluN1 LBD and glutamate binding to the GluN2A LBD drives the conformational equilibrium toward the closed LBD (*Yao et al., 2013*). While crystallographic studies have determined the binding pose of D-serine in the closed GluN1 LBD (*Furukawa and Gouaux, 2003*), the molecular mechanisms by which D-serine finds its way into and stabilizes NMDAR LBDs are not well understood.

Previous simulation studies have revealed the mechanisms by which glycine and glutamate diffuse into the LBD binding site (*Yu and Lau, 2018*). Specifically, they found that glycine binds to the GluN1 subunit by freely diffusing into the binding pocket, where it is trapped by energetically favorable interactions with key binding site residues. Glutamate, on the other hand, was found to contact residues along the protein surface that helped guide itself into its binding pocket, positioning it to interact stably with residues in the binding site. These two binding mechanisms were referred to as 'unguided' and 'guided' diffusion, respectively (*Yu et al., 2018*). This paradigm established the two extremes by which ligands enter their receptor sites: one in which stable ligand binding only depends upon the identity of the binding site residues and another that also heavily relies on residues outside the binding site to guide the ligand toward its bound pose.

Performing multi-microsecond molecular dynamics simulations of the glycosylated GluN1/GluN2A LBD dimer, we identified binding mechanisms and residues critical for promoting D-serine binding and stabilization by developing a new binding pathway clustering workflow. Surprisingly, we observed D-serine binding to both GluN1 and GluN2A LBDs. We determined that D-serine binding to GluN2A partially stabilizes the active LBD conformation. Inspired by these simulation results, we determined that D-serine competes with glutamate for binding to GluN2A via a competitive inhibition mechanism using electrophysiology measurements, where D-serine was found to be inhibitory at high concentrations. Since NMDAR LBDs are glycosylated under physiological conditions (*Kaniakova et al., 2016*), including N-linked glycans in our simulations revealed that glycans primarily regulate the binding process by stabilizing the active LBD. In total, we investigated the molecular components contributing to D-serine binding and stabilization, highlighting the complex components driving neurotransmission.

## Results

### D-serine binding pathways for GluN2A and GluN1 LBDs

In simulating the GluN1/GluN2A LBD dimer, which is a physiological NMDAR unit, we intended to focus our attention on the mechanisms by which D-serine binds to the GluN1 LBD, the subunit to which D-serine is a potent agonist. However, in our simulations, we also observed a significant number

of D-serine binding events involving the GluN2A LBD, an unexpected finding. Full binding includes both ligand association and LBD closure (*Lau and Roux, 2011*). Here, binding and unbinding refer only to ligand association and dissociation, respectively. We observe D-serine binding and unbinding multiple times throughout the trajectory (*Figure 1—source data 2*, *Figure 2—source data 1*). These binding events are primarily made up of guided-diffusion pathways in which D-serine contacts key residues on the LBD surface to help guide it into or out of the binding cleft. In our aggregate ~51 μs of sampling of the glycosylated GluN1/GluN2A LBD dimer, we identified 99 guided-diffusion pathways for GluN2A and 104 (plus 23 free diffusion events) for GluN1. Due to the stochastic nature of these pathways, we needed to develop a reliable way to identify key features of predominant binding pathways. To address this, we applied pathway similarity analysis (PSA) (*Seyler et al., 2015*) to quantify the spatial and geometric similarity between pairs of paths (*Figure 1A*, *Video 1*). Here, we extend this application to ligand binding pathways by monitoring the change in ligand $C_\alpha$ position throughout each path. This allowed us to cluster paths traversing similar regions of the LBD surface. To aid in describing the different faces of the LBD, we use an order parameter $(\xi_1, \xi_2)$ defined in previous work (*Yao et al., 2013*) to describe whether D-serine primarily contacts residues on the $\xi_1$ or $\xi_2$ face of the LBD (*Figures 1B and 2A*). For GluN2A, cluster analysis revealed four distinct regions of D-serine occupancy. The clusters correspond to the following methods of binding: 1. D-serine approaches the binding pocket from the $\xi_2$ face; 2. D-serine contacts the D1 residues on the $\xi_1$ face; 3. D-serine zigzags between D1 and D2 lobes on the $\xi_1$ face; 4. D-serine primarily contacts residues on the D2 lobe of the $\xi_1$ face (*Figure 1C–F*). Similarly, for GluN1, cluster analysis revealed four distinct clusters corresponding to similar pathways of binding: 1. D-serine contacts the $\xi_2$ face; 2. D-serine zigzags between D1 and D2 lobes on the $\xi_1$ face; 3. D-serine contacts residues on the N-terminal (top) end of D1 of the $\xi_1$ face; 4. D-serine contacts residues of D1 loop 2 that protrudes from the LBD into solution. We then analyzed the resulting clusters to identify key residues that guide D-serine into the binding site (*Figure 2B–E*, *Video 2*). Interestingly, we observed that GluN1 pathways involve fewer interactions between D-serine and D2 residues; most notably, there were fewer contacts with Helix F (Helix E for GluN2A) compared to GluN2A pathways.

To quantify the extent to which these clusters involve similar residue contacts, we used a pairwise similarity metric called the overlap coefficient (i.e., Szymkiewicz–Simpson coefficient) that describes agreement between sets of residues (*Vijaymeena and Kavitha, 2016*). Doing so provides a way to determine whether these spatial clusters are mostly made up of random contacts, or whether groups of residues tend to act together to promote binding, allowing us to quantify the extent to which agonist diffusion is 'guided' by contacts along the LBD. For GluN2A, we computed the overlap coefficient for all path pairs in each cluster for comparison with the global mean (global $\langle OC \rangle$ = 0.557) (*Figure 1—figure supplement 1A*). We found that pathway pairs in three of the four clusters yielded an overlap coefficient greater than the mean of all pairs of paths from all clusters, indicating that pathways in each cluster are made up of specific residue contacts (*Figure 1—figure supplement 1C*). In contrast, for GluN1, a significant cluster (26 paths) involving interactions with residues on the $\xi_2$ face of the LBD has a cluster mean $OC$ much less than the global mean (global $\langle OC \rangle$ = 0.671), indicating that this cluster primarily comprises random contacts (*Figure 1B*, *Figure 1—figure supplement 1B,D*, *Figure 1—figure supplement 4*). This suggests that D-serine binding to GluN1 may be more diffusion-driven and less guided than to GluN2A. Therefore, we propose that agonist binding mechanisms exist on a spectrum ranging from unguided to guided diffusion. The difference in the specificity of D-serine contacts along binding pathways for GluN2A and GluN1 suggests that the extent to which agonists rely on pathways of guiding residues depends on LBD architecture and not solely upon the identity of the agonist.

Mapping important pathway residues onto the intact GluN1/GluN2A NMDAR (PDB ID: 6MMM *Jalali-Yazdi et al., 2018*) further enriches our understanding of binding pathways by allowing us to determine whether residues in particular pathways are accessible for binding or obscured by other receptor domains and subunits. For GluN2A, access to residues on the extreme of the $\xi_2$ face is slightly restricted by the presence of the GluN1 subunit of the adjacent LBD dimer (*Figure 1—figure supplement 2A*). However, this interface does not seem to be near the specific residues identified as critical for binding. Even more restricted is access to residues on the $\xi_1$ face of GluN1, which are obscured by GluN2A of the adjacent LBD dimer, including residues identified as critical for binding pathways (*Figure 1—figure supplement 2B*). This might bias the pathways observed for the intact

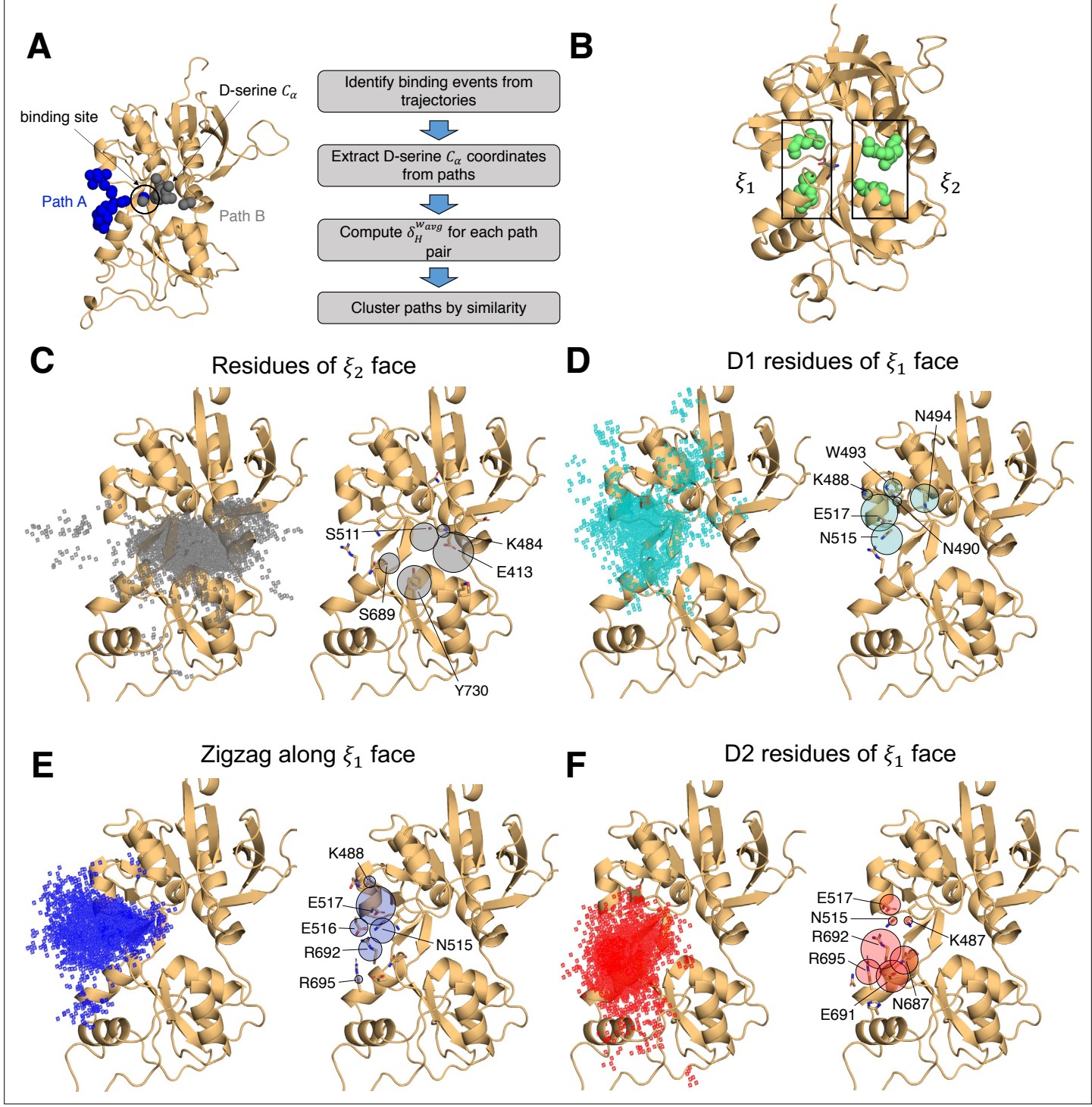

**Figure 1.** Identifying D-serine binding pathways for GluN2A using pathway similarity analysis (PSA). (**A**) Overview of the PSA workflow for quantifying similarity between D-serine binding pathways. (**B**) 2-dimensional order parameter ($\xi_1$, $\xi_2$) that describes the degree of GluN2A LBD closure. For each of the above (**C–F**), the left image shows D-serine density, while the right image shows the residues most frequently contacted by D-serine as it enters/leaves the binding site for each cluster. Labeled residues demonstrate ≥ 0.2 fractional occurrence defined relative to the most contacted residue in each cluster, but all residues with ≥ 0.1 fractional occurrence are shown in stick representation (see *Figure 1—source data 3*). (**C**) Cluster 1 involves residues of the $\xi_2$ face of the LBD. (**D**) Cluster 2 involves residues of the $\xi_1$ face of the D1 lobe. (**E**) In Cluster 3, D-serine zigzags between D1 and D2 lobe residues of the $\xi_1$ face. (**F**) Cluster 4 primarily involves D2 lobe residues on the $\xi_1$ face.

The online version of this article includes the following source data and figure supplement(s) for figure 1:

*Figure 1 continued on next page*

*Figure 1 continued*

**Source data 1.** Simulation summary: overview of simulation systems.

**Source data 2.** Record of all successful binding pathways in each simulation system for D-serine binding to GluN2A.

**Source data 3.** Per-residue contact frequency analysis for D-serine binding to GluN2A by cluster identified with PSA.

**Source data 4.** GluN2A residues most frequently contacted by D-serine given that the pathway results in successful binding – listed for each simulation system.

**Figure supplement 1.** Overlap coefficient analysis for GluN2A and GluN1 binding pathways.

**Figure supplement 2.** D-serine pathway residues mapped onto the intact GluN2A NMDAR (PDB ID: 6MMM *Hansen et al., 2021*).

**Figure supplement 3.** Degree of LBD closure during D-serine binding pathways to (**A**) GluN2A and (**B**) GluN1.

**Figure supplement 4.** Dendrograms for hierarchical clustering of weighted average Hausdorff distances for D-serine binding pathways for (**A**) GluN2A and (**B**) GluN1 according to the Ward linkage criterion.

receptor by forcing the agonist to favor residues on the $\xi_2$ face of the LBD. Since our overlap coefficient analysis of the cluster that corresponds to the $\xi_2$ face of GluN1 identified more non-specific interactions, it is possible that the D-serine mechanism would be biased to favor unguided diffusion. It is also possible that access to residues near the N-terminal end of D1 would be restricted by the R2 lobe of its own ATD.

We next investigated whether a specific LBD conformational state was favored for successful D-serine binding pathways. We computed our $(\xi_1, \xi_2)$ order parameter to quantify the degree of closure of the LBDs for all trajectory frames identified as part of binding (and unbinding) pathways and found that $(\xi_1, \xi_2) = (16,14)$ for GluN2A (*Figure 1—figure supplement 3A*) and $(\xi_1, \xi_2) = (11,13)$ for GluN1 (*Figure 1—figure supplement 3B*). These values correspond to a partially open LBD. The LBD needs to be open enough for the ligand to diffuse into the pocket but closed enough to form some stabilizing interactions with the ligand. However, we notice that the $\xi_1$ is smaller for GluN1, indicating that agonist binding can occur at slightly more closed LBD conformations. GluN1 pathways where $(\xi_1, \xi_2) = (11,13)$ are mostly in the cluster defined by D-serine interactions with Loop 2, highlighting the role of Loop 2 residues in D-serine binding to GluN1. Overall, these results suggest that the degree of LBD closure does influence the likelihood of successful binding.

## Effects of D-serine binding on the LBD conformational free energy landscapes

Since we did not expect to see D-serine binding to the GluN2A LBD, we needed to determine whether these GluN2A D-serine binding events are able to modulate the GluN2A LBD conformation. Since full LBD closure occurs on multi-microsecond to millisecond timescales (*Sinitskiy et al., 2017*; *Dolino et al., 2016*; *Rajab et al., 2021*), direct observation of such a conformational change was not fully captured from our equilibrium binding trajectories. Instead, to ensure we are sampling the full range of LBD conformations, we performed umbrella sampling free energy molecular dynamics simulations to obtain the conformational free energy landscape of GluN2A bound to D-serine (*Figure 3A*).

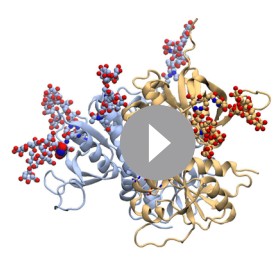

**Video 1.** Process of D-serine binding to the GluN2A LBD.

https://elifesciences.org/articles/77645/figures#video1

We used the order parameter $(\xi_1, \xi_2)$ (*Yao et al., 2013*) that captures the opening and closing motion of the LBDs observed in crystal structures of these domains. Since no crystal structure exists for D-serine bound to GluN2A, we identified residues critical for stabilizing the agonist in the closed state by analyzing contacts in lowest-energy ($\leq 1$ kcal mol$^{-1}$) conformers extracted from the 2D PMF computed from umbrella sampling simulations of D-serine bound to the GluN2A LBD (*Figure 3—figure supplement 1*). For reference, we compared the resulting energy landscape to those previously computed for the apo- and glutamate-bound GluN2A monomers (*Figure 3C and D*; *Yao et al., 2013*). In the apo PMF, there is a

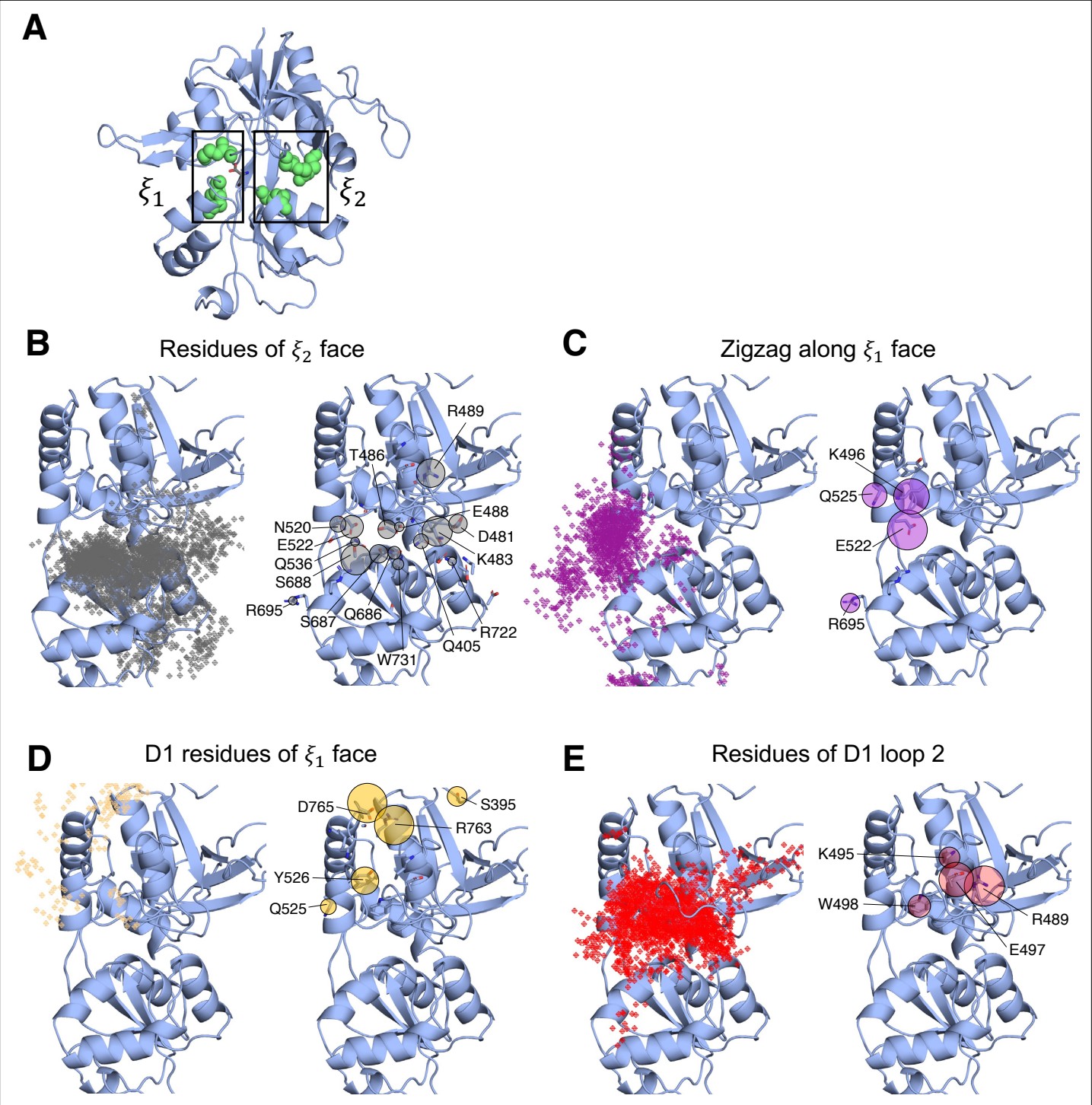

**Figure 2.** Identifying D-serine binding pathways for GluN1 using pathway similarity analysis (PSA). (**A**) 2-dimensional order parameter $(\xi_1, \xi_2)$ that describes the degree of GluN1 LBD closure. For each of the above (**B–E**), the left image shows D-serine density, while the right image shows the residues most frequently contacted by D-serine as it enters/leaves the binding site for each cluster. Labeled residues demonstrate ≥ 0.2 fractional occurrence defined relative to the most contacted residue in each cluster, but all residues with ≥ 0.1 fractional occurrence are shown in stick representation (see *Figure 2—source data 2*). (**B**) In Cluster 1, D-Serine contacts residues on the $\xi_2$ face of the LBD. (**C**) Cluster 2 involves interactions with both D1 and D2 residues of the $\xi_1$ face. (**D**) Cluster 3 involves contacts with residues at the top of the D1 lobe on the $\xi_1$ face. (**E**) Cluster 4 is defined by interactions with D1 loop 2 that reaches into solution.

The online version of this article includes the following source data for figure 2:

*Figure 2 continued on next page*

*Figure 2 continued*

**Source data 1.** Record of all successful binding pathways in each simulation system for D-serine binding to GluN1.

**Source data 2.** Per residue contact frequency analysis for D-serine binding to GluN1 by cluster identified with PSA.

**Source data 3.** GluN1 residues most frequently contacted by D-serine given that the pathway results in successful binding – listed for each simulation system.

wide energy minimum that accommodates more open LBD conformations; in contrast, the glutamate-bound PMF exhibits a narrow and steep energy minimum at the closed state. We see that, like glutamate, D-serine stabilizes the closed LBD conformation. The D-serine energy landscape has a global minimum corresponding to $(\xi_1, \xi_2)$ values of (11, 11.5 Å) and a metastable minimum corresponding to $(\xi_1, \xi_2)$ values of (15.5, 11.5 Å). The presence of a metastable agonist-bound LBD partially open intermediate suggests that D-serine may not stabilize the closed conformation to the same extent as glutamate and generate sufficient force to control channel gating. We then compared different conformers corresponding to these two states to determine residues critical for agonist stabilization. The primary difference between the residue contacts in conformers of the two states is the prevalence of interactions with Thr-690 (*Figure 3—figure supplement 1B*), which only contacts D-serine in the more closed state centered at $(\xi_1, \xi_2)$ = (11, 11.5 Å). This is supported by our binding simulations; although we do not fully sample LBD closure, trajectory frames with low $(\xi_1, \xi_2)$ values involve contacts with Thr-690. This suggests that Thr-690 is critically involved in promoting full GluN2A LBD closure upon agonist binding.

Experimental binding studies have indicated that D-serine may be a more potent GluN1 agonist than glycine (*Mustafa et al., 2004*). To better understand the molecular mechanism responsible for this difference in agonist potency, we computed the conformational free energy for the D-serine-bound GluN1 LBD (*Figure 3F*). Compared with the previously computed glycine-bound and apo LBDs (*Figure 3G and H*; *Yao et al., 2013*), the presence of D-serine in the binding cleft results in a greater population of conformers in the closed conformation and fewer conformers adopting a more open conformation. Similar to GluN2A Thr-690, GluN1 Asp-732 and (to a lesser extent) Ser-688 help stabilize D-serine in the closed LBD conformation by interacting with the D-serine hydroxyl. For this reason, we propose that D-serine's high potency is due, at least in part, to its ability to more strongly stabilize a closed LBD through additional interactions with the D2 lobe.

## D-serine and glutamate compete for binding to the GluN2A LBD

Since our simulations revealed that D-serine can enter the GluN2A LBD binding pocket and partially stabilize the active conformation, we hypothesized that D-serine might compete with glutamate for binding to GluN2A. In fact, we observed D-serine binding to GluN2A, even in the presence of glutamate, although glutamate bound more frequently and with longer residence times in the binding site (*Figure 1—source data 2*, *Figure 5—source data 1*). Specifically, there are 17 successful associations for glutamate in the 15 µs glycosylated mixed-agonist trajectory compared with 5 for D-serine, and 75 glutamate binding events compared with 6 D-serine binding events for the non-glycosylated mixed-agonist trajectory. In addition, the average time bound for glutamate was 131 ns (glycosylated) and 236 ns (non-glcyolated) compared with 3 ns (glycosylated) and 56 ns (non-glycosylated) for D-serine. Since increasing the D-serine concentration would increase the frequency of D-serine binding to GluN2A, it is possible that D-serine could function as an inhibitor (competitive antagonist) at high concentrations.

To probe this behavior experimentally, we measured GluN1-2A NMDAR currents using two-electrode voltage clamp (TEVC) electrophysiology. We observed that at high (~1 mM) D-serine concentrations, NMDAR activity was inhibited (*Figure 4A*). The inhibition was dependent

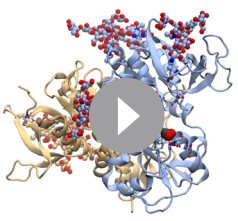

**Video 2.** Process of D-serine binding to the GluN1 LBD.

https://elifesciences.org/articles/77645/figures#video2

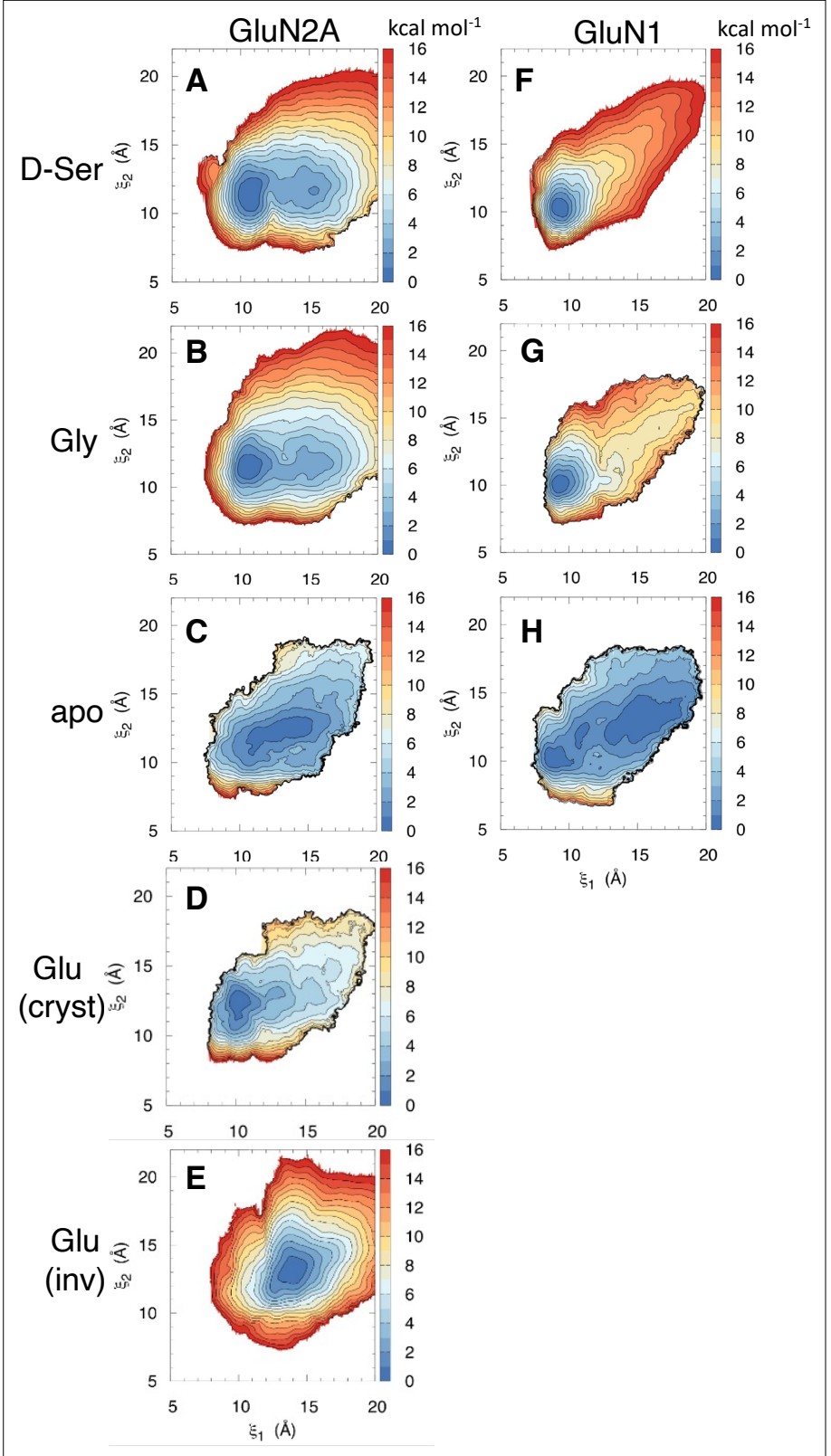

**Figure 3.** Conformational free energy landscapes for GluN2A and GluN1 LBDs. Umbrella sampling molecular dynamics simulations were used to compute the potential of mean force (PMF) along the $(\xi_1, \xi_2)$ order parameter for (**A**) D-serine bound to GluN2A, (**B**) glycine bound to GluN2A, (**C**) apo GluN2A previously computed in [196], (**D**) glutamate bound to GluN2A in its crystallographic pose previously computed in [196], (**E**) glutamate bound

*Figure 3 continued on next page*

*Figure 3 continued*

to GluN2A in the inverted pose identified in [218], (**F**) D-serine bound to GluN1, (**G**) glycine bound to GluN1 previously computed in [196], (**H**) apo GluN1 previously computed in *Yao et al., 2013*.

The online version of this article includes the following source data and figure supplement(s) for figure 3:

**Source data 1.** Per-residue contact frequency analysis of the bound state for each agonist computed from lowest-energy conformers extracted from umbrella sampling simulations.

**Figure supplement 1.** GluN2A residues contacting D-serine in lowest-energy conformers.

**Figure supplement 2.** Error of umbrella sampling PMFs computed by block averaging for (**A**) D-serine bound to GluN2A, (**B**) glycine bound to GluN2A, (**C**) glutamate bound to GluN2A in the inverted pose, and (**D**) D-serine bound to GluN1.

on glutamate concentrations, implying that the inhibitory effect of D-serine may be competitive (*Figure 4B*). Furthermore, dose-response curves of glutamate activation were right-shifted in the presence of increasing concentrations of D-serine (*Figure 4C*). The calculated slope value of the Schild plot at 1.1 ± 0.1 implied that D-serine and glutamate likely compete against each other (*Figure 4C*). Combined with our simulation results, our electrophysiological data support the hypothesis that D-serine at high concentrations can bind to the GluN2A subunit and compete against glutamate.

Since a similar inhibitory effect was also observed at high glycine concentrations by TEVC electrophysiology (*Figure 4A*), we repeated our umbrella sampling simulations with glycine bound to the GluN2A LBD. We see that glycine also favors the closed LBD (*Figure 3B*). The lowest-energy conformers of GluN2A with glycine are fastened shut by contacts between the N-terminal amine of glycine and Tyr-730. Although glutamate still stabilizes the closed GluN2A LBD to the greatest extent, comparable thermodynamics between different agonists suggest that kinetics of agonist binding and unbinding is a critical driver of agonist-induced activation. The GluN2A LBD likely never closes around glycine because glycine does not remain bound long enough to induce LBD closure.

Previous binding studies *Mayer, 2017* have indicated that glutamate, the primary GluN2A agonist, similarly relies on LBD surface residues to promote binding. To determine whether D-serine and glutamate binding are guided by similar residue contacts, we computed the overlap coefficient between residues in D-serine and glutamate pathways to be 0.964 for the glycosylated GluN2A LBD, corresponding to a significant overlap in agonist occupancy (*Figure 5A*). This high degree of overlap between glutamate and D-serine pathway residues indicates that they bind through similar mechanisms. To assess the importance of pathway residues for D-serine binding to the LBD dimer, we performed TEVC electrophysiology to obtain D-serine dose-response curves for various pathway mutants for GluN1 and GluN2A (*Figure 4D*). Notably, the GluN2A mutants Arg692Ala and Arg695Ala showed two to three-fold decreased D-serine inhibition potency. This result suggests that these two GluN2A residues play a role in D-serine inhibition and D-serine guided-diffusion pathways. Since these residues are also involved in glutamate binding pathways, this finding more generally supports the guided-diffusion mechanism by which agonists bind to GluN2A. The absence of this effect on the two GluN1 pathway mutants supports the increased diffusive behavior of D-serine binding to GluN1.

To determine the extent to which D-serine and glutamate binding are guided by similar residue contacts, we computed the overlap coefficient between residues in D-serine and glutamate pathways to be 0.964 for the glycosylated GluN2A LBD, corresponding to a significant overlap in agonist occupancy (*Figure 5A*). This high degree of overlap between glutamate and D-serine pathway residues indicates that they bind through similar mechanisms.

Despite similar pathway residues, we identified key residues that distinguish glutamate from D-serine binding pathways (*Figure 5B* and *Figure 5—source data 2*). Most of the residues important for D-serine binding, but not for glutamate binding, are located on the $\xi_2$ face of the LBD. Most notably, Glu-413, Tyr-730, Ser-511, and Asp-731 all occur in D-serine binding pathways with a frequency of more than ten times their fractional occurrence in glutamate binding pathways. Due to the locations of these residues (involved in either LBD closure or dimerization), we were unable to experimentally assess their effect on the D-serine inhibition. It is important to note, however, that glutamate does interact with residues on the $\xi_2$ face, but the specific nature of those contacts differs between the two agonists. In contrast, we found that Lys-487 is contacted with significantly greater frequency in glutamate binding pathways. Due to these residues' close proximity to the binding cleft,

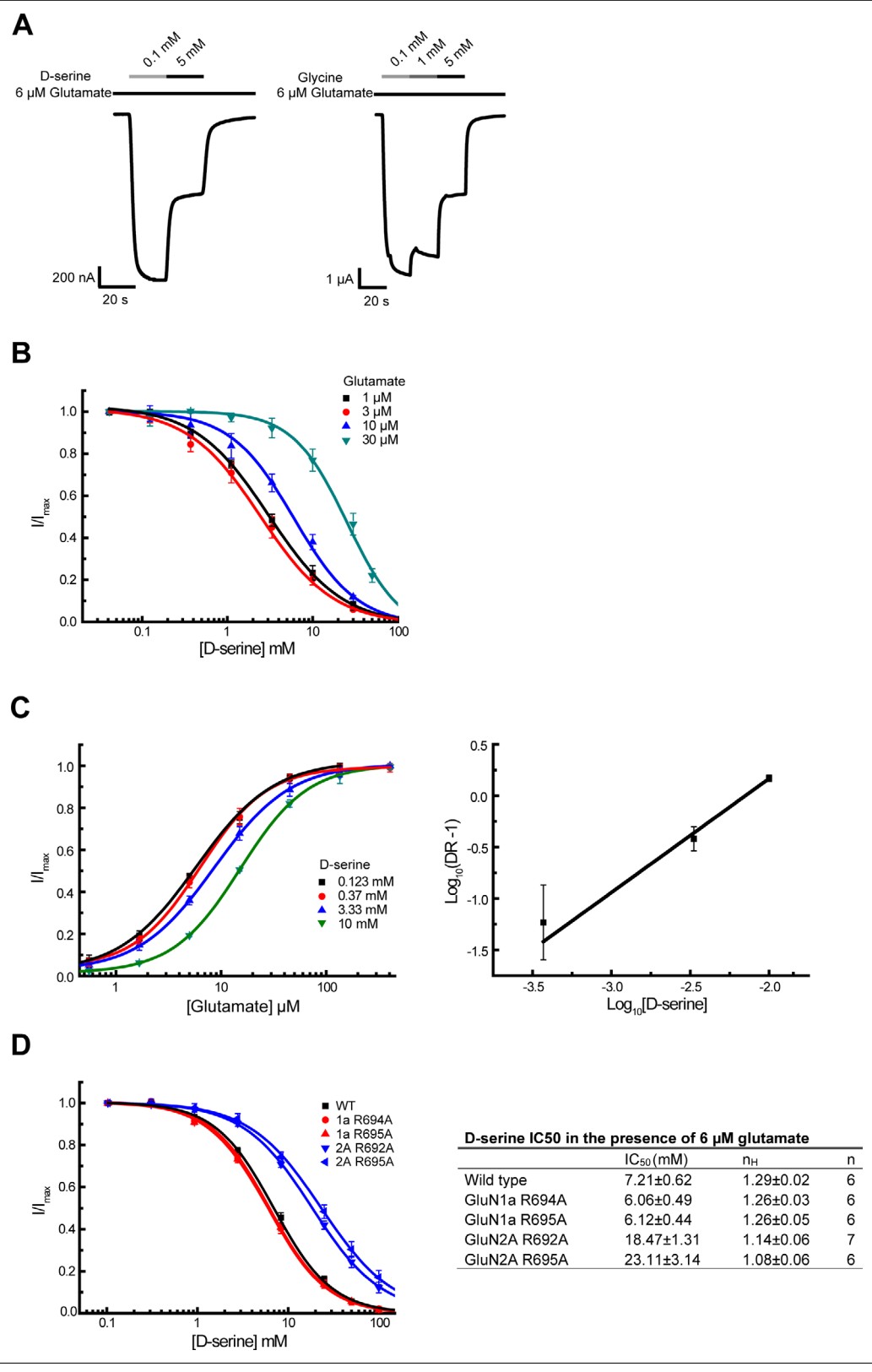

**Figure 4.** D-serine competes glutamate binding as an antagonist at high concentration. (**A**) Representative Two-electrode voltage clamp (TEVC) recording on GluN1/GluN2A NMDARs expressing oocytes. The traces show inhibition of the NMDAR current by the GluN1 agonists D-serine (left) and glycine (right) at a high concentration. 6 µM of glutamate is present throughout the recording. (**B**) D-serine inhibition at various concentrations of glutamate

*Figure 4 continued on next page*

*Figure 4 continued*

(1, 3, 10, and 30 µM). . (**C**) Glutamate responses at various concentrations of D-serine (0.123, 0.37, 3.33 and 10 mM) (left). Schild plot analysis of D-serine competition against glutamate (right). The calculated slope of the Schild plot was 1.11 ± 0.13 and the intercept was 2.38 ± 0.26. DR stands for dose ratio. (**D**) D-serine inhibition curves (left) and $IC_{50}$ values for various pathway residue mutants on GluN1 and GluN2A LBDs (right). The pairwise comparison shows that the changes in $IC_{50}$ values of the mutants from the wild type are significant. The statistical analysis was done by two-tail t-test where the p values are GluN1a-R694A = 0.0061, GluN1a-R695A = 0.0065, GluN2A-R692A = $4.1 \times 10^{-9}$, and GluN2A-R695A = $3.3 \times 10^{-5}$. All experiments were repeated in at least four independent oocytes. Error bars represent the average current ± SD.

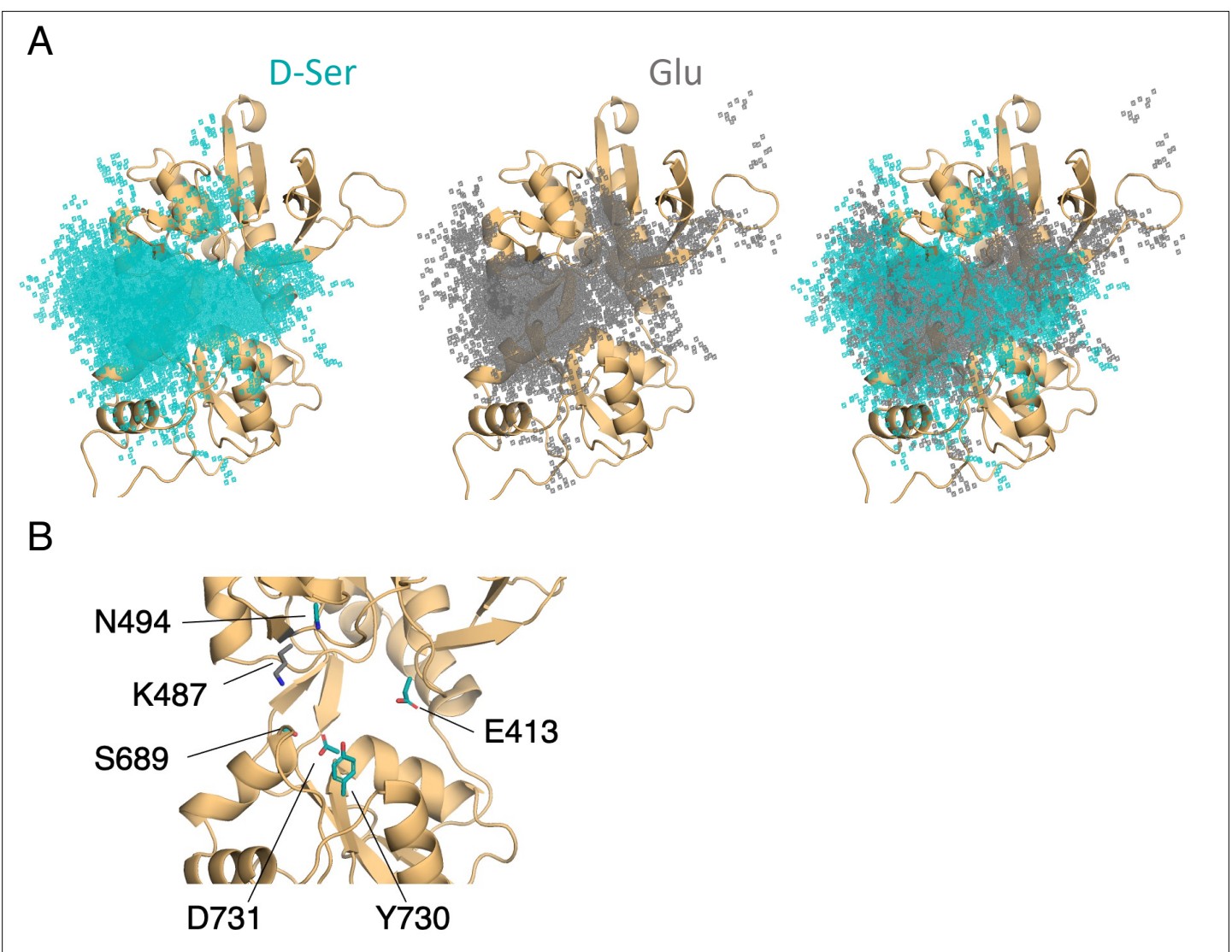

**Figure 5.** Comparison of D-serine and glutamate binding to GluN2A. (**A**) Overlay of D-serine (teal) and glutamate (gray) density. (**B**) Residues that distinguish D-serine (teal) from glutamate (gray) binding pathways (see *Figure 5—source data 2*).

The online version of this article includes the following source data for figure 5:

**Source data 1.** Record of all successful binding pathways in each simulation system for glutamate binding to GluN2A.

**Source data 2.** Comparison of relative residue contact frequency for D-serine and glutamate.

it is likely that these residues are responsible for facilitating proper positioning of the agonists in the binding site, based on differences in agonist size and shape.

An important feature of glutamate binding to GluN2A is its ability to bind in an inverted pose relative to the crystal structure, which we observed in previous simulations (*Yu and Lau, 2018*; *Yu et al., 2018*). Since no experimental structure exists for glutamate bound in the inverted pose, we performed umbrella sampling simulations to determine the free energy landscape of the GluN2A LBD with glutamate bound in the inverted pose (*Figure 3E*). We found that glutamate bound in the inverted pose prevents full LBD closure as predicted in previous work (*Yu and Lau, 2018*). Specifically, glutamate in the inverted pose stabilizes a conformation centered around $(\xi_1, \xi_2)$ values of (14, 13 Å). Comparing the low-energy conformers of D-serine and inverted glutamate ($\leq 1$ kcal mol$^{-1}$) with the glutamate-bound crystal structure, we found that D-serine and glutamate are stabilized by the same residues, although there are fewer interactions between Thr-690 and glutamate in the inverted pose, further supporting the importance of this residue for stabilizing the fully closed LBD.

## Kinetic analysis of D-serine binding pathways

We computed the D-serine association rate constant ($k_{on}$) for GluN2A and GluN1 LBDs using a method described (*Dror et al., 2011*) and used in previous iGluR work (*Yu et al., 2018*) as summarized in the equation below:

$$k_{\mathrm{on}} = \frac{N_b}{\sum_i \frac{t_i [L_i]}{s_i}}.$$

Here, $N_b$ is the number of association events, $t_i$ is the time the agonist spends in bulk solvent, $s_i$ is the number of identical binding sites, and $[L_i]$ is the concentration of free agonist. One advantage of this approach is the ability to combine simulations performed at various concentrations of free agonist $[L_i]$. Here, $k_{on}$ is a bulk property and relies on fully sampling the LBD conformational landscape throughout the simulation. However, our binding simulations fail to adequately sample the agonist-bound, closed LBD state. This affects both the number of observed binding events $N_b$ and the time the agonist spends in bulk solvent ($t_i$). Since this value is most sensitive to the number of identified binding events $N_b$, we computed the $k_{on}$ for different $N_b$ values based on the duration of the resulting binding event. This minimizes contributions from extremely short binding events that are unlikely to be functionally relevant. For GluN2A, this results in a D-serine $k_{on}$ with an upper bound of $7.8 \times 10^7$ M$^{-1}$s$^{-1}$ (all events included) and a lower bound of $1.6 \times 10^7$ M$^{-1}$s$^{-1}$ (only events with agonist residence times >100 ns were included). For GluN1, the upper bound for $k_{on}$ is $9.0 \times 10^7$ M$^{-1}$s$^{-1}$ and the lower bound is $7.0 \times 10^6$ M$^{-1}$s$^{-1}$. Based on these values, it is reasonable to expect that D-serine binds to GluN2A and GluN1 at similar rates. For comparison, the association rate constants computed for glutamate binding to GluN2A with this method range from $4.9 \times 10^7$ M$^{-1}$s$^{-1}$ to $1.4 \times 10^8$ M$^{-1}$s$^{-1}$. Similar ranges of D-serine binding rate constants for GluN2A and GluN1 support our data indicating a guided-diffusion mechanism. However, this definition of the association rate constant does not capture the molecular details that produce this bulk behavior.

For agonist binding mechanisms dominated by guided diffusion, we can monitor how much time the agonist spends (1) in bulk solvent, (2) associated with the LBDs, and (3) docked in the binding cleft (interacting with the conserved arginines Arg-523 for GluN1 or Arg-518 for GluN2A). Transitions between these states can be represented by the following three-step process:

$$P + L \rightleftharpoons PL_{\mathrm{assoc}} \rightleftharpoons PL_{\mathrm{docked}}.$$

Here, the $PL_{assoc}$ state either results in successful binding (represented by pathways) or nonspecific interactions resulting in dissociation. From the clusters of residues that we identified in our pathway similarity analysis, we determined to what extent a particular residue is critical for guiding the agonist into the binding site using a conditional probability-based framework (*Figure 1—source data 4*, *Figure 2—source data 3*). For GluN2A, given that a binding event results in successful agonist docking, residues Asp-515, Glu-517, Arg-692, Asn-687, Lys-487, Lys-484, and Ser-689, Lys-488, Ser-511, and Glu-413 are contacted most frequently across all datasets. Given successful D-serine binding, contacts with GluN1 residues Lys-496, Lys-495, Trp-498, Arg-489, and Glu-497 occur in the greatest number of pathways. Slightly less agreement in crucial GluN1 binding residues across datasets further supports a more diffusive/random binding mechanism for D-serine binding to GluN1.

Calculating the number of successful binding events compared with random associations allows us to determine the level of noise present in the binding process. For glycosylated simulations with 19.6 mM D-serine, we observe an average of 1242 ± 31 random GluN2A associations (n=3 simulations) per microsecond (1249 ± 1 for GluN1, n=2 simulations) that fail to result in successful binding. In these same simulations, we observe about 1.2 ± 0.6 successful GluN2A binding events per microsecond (1.0 ± 0.5 for GluN1).

Supporting our guided-diffusion mechanism, we identified residues for which the ratio of successful to random binding was increased. In general, GluN2A LBD residues contacted by the agonist experience 26 ± 2 random associations per microsecond (31 ± 1 for GluN1). For each residue involved in successful D-serine binding pathways, we calculated the percentage of associations resulting in successful binding. For residues important for guided-diffusion pathways, this percentage is >1% (*Figure 1—source data 4*, *Figure 2—source data 3*). This allows us to quantify the importance of pathway residues despite a noisy non-specific association signal.

## Role of N-linked glycans in D-serine binding pathways

In addition to identifying residues that are responsible for agonist specificity in binding pathways, we also explored the effect of the N-linked $Man_5GlcNAc_2$ (Man5) glycans (*Figure 6—figure supplement 1A*) on the residues involved in agonist binding pathways. Previous electrophysiological studies have indicated that glycans function as LBD potentiators (*Sinitskiy and Pande, 2017*). In our simulations, we observed that near-pocket glycans appear to 'reach' into the binding pocket. This reaching behavior was observed in previous simulations of the glycosylated NMDAR LBDs in which the glycan forms a 'cage' around the binding pocket by forming interactions with the LBD D2 lobe and is believed to be associated with NMDAR potentiation by glycans (*Sinitskiy and Pande, 2017*). For GluN2A, there are two glycans that are near the binding pocket: N443-Man5 and N444-Man5, both of which can interact with the LBD D2 lobe (*Figure 6A*). For GluN1, there is a single glycan N491-Man5 that adopts this caged conformation (*Figure 6B*). To quantify this behavior in our simulations, we developed a general order parameter to describe the relationship between the glycan and the LBD D2 lobe that measured the minimum distance between any glycan heavy atom and any residue on the LBD D2 lobe. From this order parameter, we computed glycan PMFs along the glycan-D2 order parameter for each near-pocket glycan (*Figure 6C–E*).

We compared our glycosylated trajectories with an additional 30 μs of simulations of the non-glycosylated GluN1/GluN2A LBD dimer to identify ways in which the presence of glycans influences binding pathways. Our data indicate that residues on the $\xi_2$ face are contacted more frequently in non-glycosylated simulations, although these residues are important for D-serine binding with and without glycans (*Figure 6—source data 1*). GluN2A residues Asp-515 and Glu-517, are contacted more frequently in glycosylated systems. The frequency with which D-serine interacts with GluN1 residue Arg-489 in pathways is greater for glycosylated pathways than those without glycans. On average, glycan-mediated D-serine interactions result in slightly longer pathways, suggesting that the presence of glycans slows down the binding process, setting up small kinetic 'traps'.

When we analyzed glycan behavior in our binding pathways, we found that very few D-serine binding pathways (27% for both GluN2A and GluN1) involve contacts with glycans. While glycan-agonist interactions make up a small percentage of time spent in binding pathways (10% for GluN2A and GluN1 D-serine pathways), patterns in glycan interactions with the agonist as it binds suggest that glycans contribute to binding pathways in a consistent way. The most common glycan-mediated D-serine-LBD interactions for GluN2A involve an interaction network formed by N443-Man5 with Glu-412, Lys-438 (*Figure 6—figure supplement 1B*), Lys-738, Glu-413 (*Figure 6—figure supplement 1C*), Tyr-730, and Ser-511 (*Figure 6—figure supplement 1D*), as D-serine moves into the binding pocket. Another contact network formed by N444-Man5 with Lys-487, Asn-687 (*Figure 6—figure supplement 1E*), Arg-692, Arg-695, (*Figure 6—figure supplement 1F*), and Glu-413 (alongside the N443-Man5 glycan). For GluN1, the N491-Man5 glycan interacts with D-serine, trapping it in a network of interactions dominated by Arg-489 (*Figure 6—figure supplement 1G*). When formed, this contact network functions as a kinetic trap that results in longer binding pathways. Additionally, the N440-Man5 glycan also contacts D-serine as it interacts with Arg-489 and Glu-497 (*Figure 6—figure supplement 1H*). It is interesting to note that, unlike the glycan-mediated contacts identified for GluN2A, glycan-mediated agonist contacts for GluN1 do not involve D2 lobe residues. These

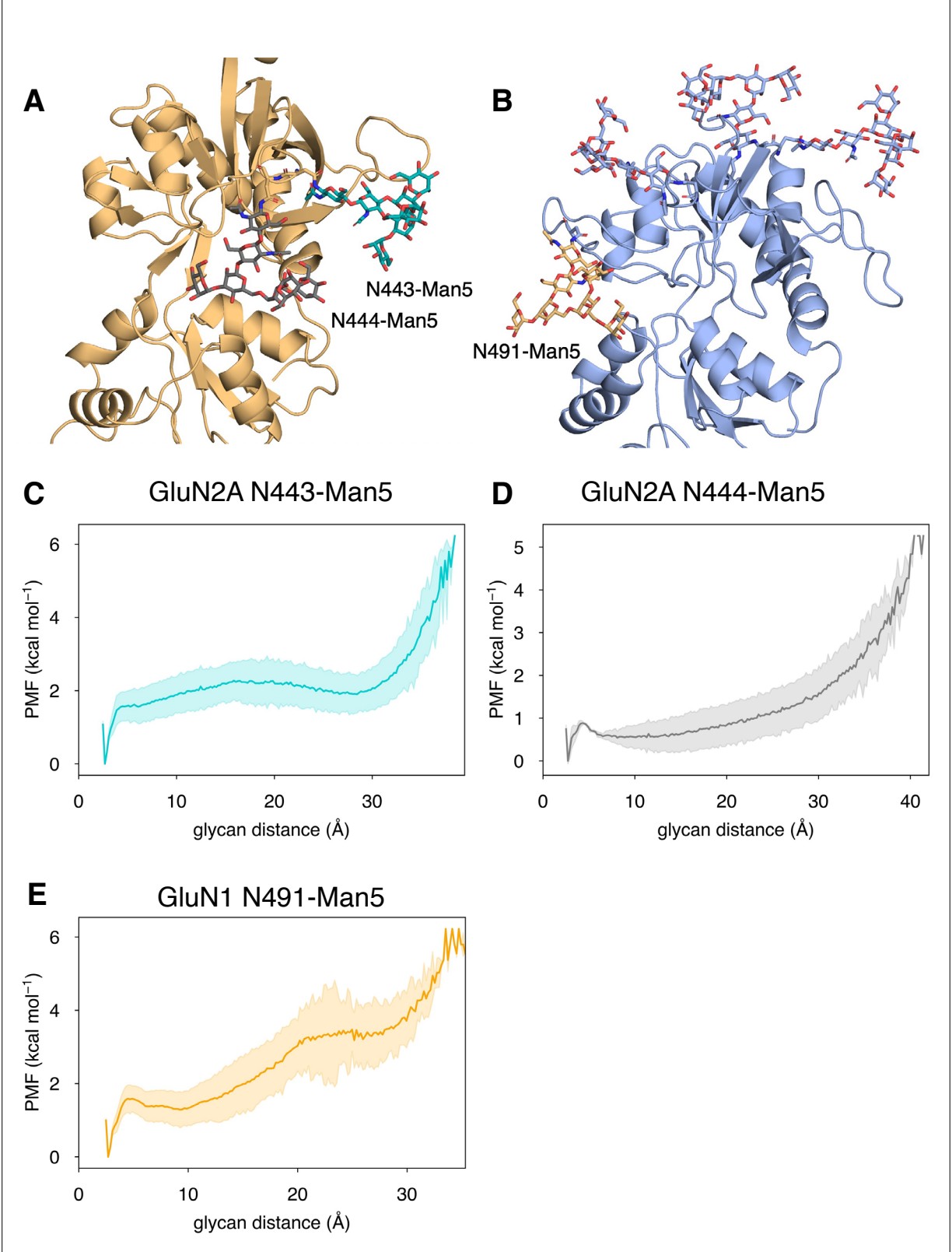

**Figure 6.** Conformational dynamics of near-pocket glycans. N-linked Man$_5$GlcNAc$_2$ (Man5) glycans (**A**) N443-Man5 and N444-Man5 for GluN2A and (**B**) N491-Man5 for GluN1. Glycan conformational energy landscapes for (**C**) GluN2A N443-Man5, (**D**) GluN2A N444-Man5, and (**E**) GluN1 N491-Man5 were obtained by computing the minimum distance between all glycan heavy atoms and D2 lobe residues and binning the distribution from all glycosylated simulation systems. Shaded error regions were computed using a block-averaging scheme described in Methods.

*Figure 6 continued on next page*

*Figure 6 continued*

The online version of this article includes the following source data and figure supplement(s) for figure 6:

**Source data 1.** Comparison of relative residue contact frequency during GluN2A and GluN1 binding pathways for glycosylated and non-glycosylated simulations.

**Figure supplement 1.** N-linked $Man_5GlcNAc_2$ (Man5) glycans interacts with D-serine as it binds.

**Figure supplement 2.** Glycan-D2 distance dependence on agonist binding for the (**A**) GluN2A N443-Man5 glycan, (**B**) GluN2A N444-Man5 glycan, and (**C**) GluN1 N491-Man5 glycan.

glycan-mediated interactions illustrate how glycan conformation can play a functional role through involvement with agonist binding and LBD conformational dynamics. However, since glycan-mediated interactions are so infrequent, the potentiating effect of glycan-D2 interactions dominates functionally.

We quantified the dependence of glycan conformation on agonist binding and LBD conformation by comparing glycan PMFs for different LBD conformations. For GluN2A, we found that glycan-D2 interactions occur more readily when the LBD is closed (calculated using a 1-dimensional projection of our LBD order parameter $\xi_{12}$, see Methods). This effect was more dramatic for N443-Man5 than for N444-Man5 (*Figure 7A and B*). A similar relationship was determined for the N491-Man5 glycan of GluN1 (*Figure 7C*); this is consistent with previous simulations (*Sinitskiy and Pande, 2017*) that suggest that N491-Man5 acts as a latch that stabilizes LBD closure. No significant relationship

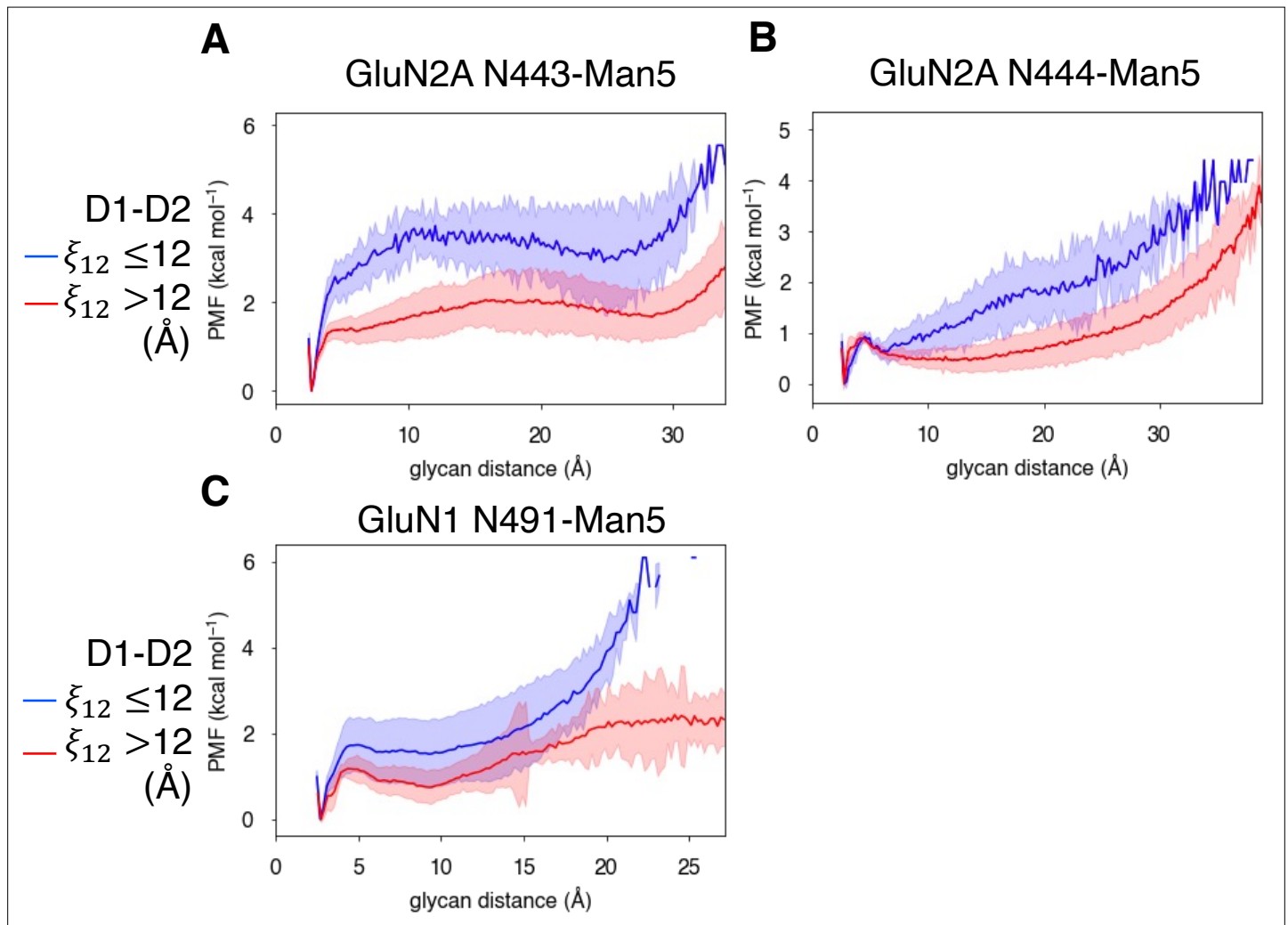

**Figure 7.** Glycan-D2 distance dependence on LBD closure for the (**A**) GluN2A N443-Man5 glycan, (**B**) GluN2A N444-Man5 glycan, and (**C**) GluN1 N491-Man5 glycan.

between glycan-D2 distance and the presence of an agonist (D-serine, glutamate, or both) in the binding site was observed (*Figure 6—figure supplement 2A-C*).

## Discussion

Here, we characterized the guided-diffusion mechanism that drives D-serine binding to NMDAR LBDs. Instead of binding solely to the GluN1 LBD, we observed substantial D-serine binding to the GluN2A LBD, a subunit widely accepted to bind to the neurotransmitter glutamate. We showed by electrophysiology that D-serine at high concentration can compete against glutamate at GluN2A, which in turn inhibits the channel activity. In the context of synaptic transmission, our finding implies that D-serine could play a role in modulating the strength of synaptic transmission. The synaptic concentration of glutamate ranges from nanomolar concentrations *Chiu and Jahr, 2017* to >1 mM following an action potential (*Dzubay and Jahr, 1999*). The synaptic concentration of D-serine is unclear, however; the extracellular concentration of D-serine ranges from 5 to 7 µM (*Matsui et al., 1995*; *Hashimoto et al., 1995*). Possible routes for D-serine to enter the synapse include vesicular release by astroglia (*Mothet et al., 2005*) and transport by Asc-1 (*Sason et al., 2017*).

Free energy landscapes computed for GluN2A bound to glutamate (*Yao et al., 2013*), D-serine, and glycine all indicate stabilization of the closed LBD bi-lobe, which is the conformational state required for receptor activation. Agonists that can interact extensively with bottom-lobe residues stabilize this state. Since glutamate does this to the greatest extent, it is likely that D-serine does not generate sufficient force to fully gate the ion channel. Subtle differences in the thermodynamics of agonist stabilization suggest that kinetics further distinguish individual agonists. While glutamate has a slightly higher association rate than D-serine, differences between association rates across agonists and subunits are not drastic. We hypothesize that, in order for agonist binding to result in NMDAR activation, the agonist must remain in the binding site long enough to induce closure—we found that this is largely dependent upon the number and strength of stable contacts the agonist forms with both D1 and D2 lobe residues.

We determined the role of N-linked glycans in agonist binding and stabilization. Glycans impact agonist binding kinetics less by direct glycan-agonist interactions and more by stabilizing the closed LBD through glycan-D2 interactions. This bias toward LBD closure would increase the agonist residence time and potentiate NMDAR activity.

Our adaptation of pathway similarity analysis allowed us to identify clusters of residues critical for binding agonists. This also allowed us to determine that the presence of pathways depends on the degree of LBD closure. We also observed that D-serine binds to GluN2A using similar pathways and residues as glutamate, while the locations of key D-serine cluster residues for GluN1 are different. Applied more broadly to drug-binding simulations, this method of analyzing binding pathways provides a useful framework for gleaning biological insight from noisy and diffusive binding data.

## Methods
### Equilibrium molecular dynamics simulations

A construct of the GluN1/GluN2A dimer based on crystal structure PDB ID: 2A5T *Furukawa et al., 2005* used in our previous study (*Yu and Lau, 2018*) was used as a starting model. The residue numberings are based on the Uniprot numbering for GRIN1 and GRIN2a entries. $Man_5GlcNAc_2$ (Man5) glycans were added using CHARMM-GUI *Glycan Reader & Modeler* (*Jo et al., 2008*; *Jo et al., 2011*; *Park et al., 2017*; *Park et al., 2019*) to asparagine residues 440, 471, 491, and 771 of GluN1 and asparagine residues 443 and 444 of GluN2A in accordance with physiologically relevant glycosylation sites (*Kaniakova et al., 2016*). GluN2A was chosen as the GluN2 subtype both to facilitate comparison with previous simulation studies and because recent evidence has suggested that the GluN2A subtype is the primary subtype at synapses, where D-serine is the dominant co-agonist (*Papouin et al., 2012*).

All systems were solvated in a 140 Å ×110 Å ×110 Å orthorhombic water box with ~150 mM NaCl using CHARMM (*Brooks et al., 2009*). All systems were electrically neutral. All simulations in this work were performed using the CHARMM36 forcefield (*MacKerell et al., 1998*) and TIP3P water model (*Jorgensen et al., 1983*). The systems were pre-equilibrated using NAMD 2.13 (*Phillips et al.,*

*2005*) first using NVT conditions and gradually relaxing backbone-sidechain restraints and then for 15 ns using NPT conditions at a pressure of 1 atm and a temperature of 310 K. The pre-equilibrated systems were then simulated on Anton 2 provided by the Pittsburgh Supercomputer Center (*Shaw et al., 2014*). A weak center-of-mass restraint of 0.5 kcal mol⁻¹ Å⁻² was applied to GluN2A N, CA, and C atoms of residues 461–463. 507–509, and 523–525 to prevent large protein translational motion. Simulations on Anton 2 were carried out at 310 K with the NPT ensemble and with the weak center-of-mass restraint of 0.3 kcal mol⁻¹ Å⁻² in accordance with previous simulations (*Yu et al., 2018*). Additional simulation details are provided in *Figure 1—source data 1*.

## Identification of binding pathways

Identifying frames in which the ligand is bound in the receptor's binding pocket provides key information about the ligand's binding affinity and the bound ensemble; however, it fails to account for the process by which the ligand enters and leaves the binding pocket. In guided diffusion, the residues that guide the ligand into the binding pocket are critical for promoting the bound state. While imposing a simple distance cutoff is sufficient for identifying the fully bound state, identifying the pathways by which the ligand binds is less trivial. Here, we introduce a 'binding chains' paradigm for defining the ligand's path along the protein. These binding chains are defined from ligand association to dissociation. An association begins when any polar ligand heavy atom comes within 6 Å of any protein polar heavy atom. The ligand is considered associated until it diffuses beyond 10 Å from the protein. The resulting chains are then filtered by contact with the selected 'docking' residue(s). Here, we use the conserved arginine residue for each subunit (Arg-523 for GluN1 and Arg-518 for GluN2A) as the essential docking residue. These chains are filtered then split into their 'binding' and 'unbinding' components by a more specific docking criterion. In our case, we require that the NH1 and NH2 atoms of the conserved arginine be within 4 Å of the ligand carboxyl in accordance with the following scheme:

- Condition 1: Arg NH1 is within 4 Å of the ligand OT1 <u>AND</u> Arg NH2 is within 4 Å of the ligand OT2

OR

- Condition 2: Arg NH2 is within 4 Å of the ligand OT1 <u>AND</u> Arg NH1 is within 4 Å of the ligand OT2

This scheme accounts for both the crystallographic binding pose (Condition 2) and a 'flipped' ligand orientation (Condition 1). Chains that fail to meet these criteria are discarded. Since binding and unbinding pathways can be considered reversible, we combine them in our analysis, reversing the order of the unbinding pathways so that all pathways have the same directionality. This results in a series of binding pathways we can characterize both geometrically and in terms of key residue interactions.

## Pathway similarity analysis and clustering

Pathway similarity analysis (PSA) was applied to each binding pathway by monitoring the agonist position as it binds. PSA involves computing a pairwise distance metric between paths that serves as a measure of geometric similarity (*Seyler et al., 2015*). The weighted average Hausdorff distance was selected as the path metric because it gave the most geospatially distinct clusters of agonist density around the protein. This weighted average Hausdorff distance was computed for all pairs of paths using the following formula as described in previous work (*Seyler et al., 2015*) and implemented in the MDAnalysis python package (*Michaud-Agrawal et al., 2011*; *Gowers et al., 2016*). The weighted-average Hausdorff distance between two paths $A$ and $B$ can be expressed as:

$$\delta_H^{w_{avg}}(A, B) = \frac{1}{2}\left[\frac{1}{|A|}\delta_H^{\text{sum}}(A|B) + \frac{1}{|B|}\delta_H^{\text{sum}}(B|A)\right],$$

where $|A|$ and $|B|$ are the number of frames in paths $A$ and $B$, respectively, and $\delta_H^{\text{sum}}$ is the one-sided summed Hausdorff distance from path $A$ to path $B$,

$$\delta_H^{\text{sum}}(A|B) = \sum_{a \in A} \min_{b \in B} d(a, b).$$

Here, $d(a,b)$ represents the distance between point $a$ of path $A$ and point $b$ in path $B$. For our system, each point $a$ is the agonist $C_\alpha$ position for a single frame in path $A$, and each point $b$ is the agonist $C_\alpha$ position for a single frame in path $B$. Therefore, $d(a,b)$ represents the Euclidean distance between the agonist $C_\alpha$'s of points in paths $A$ and $B$. $\delta_H^{\text{sum}}(A|B)$ is then computed by summing the shortest distance from each point $a$ in path $A$ to any point $b$ of path $B$ overall points in path $A$. Each of the normalized one-sided sums $\delta_H^{\text{sum}}(A|B)$ and $\delta_H^{\text{sum}}(B|A)$ is then averaged with equal weights. This does not give more weight to pathways with more frames, thus removing the temporal component from the analysis. Temporal patterns in binding pathways are analyzed for the spatial clusters separately.

These path pairs were then clustered using hierarchical clustering according to their weighted-average Hausdorff distances with the Ward (minimum variance) linkage criterion as described in previous work (*Seyler et al., 2015*) and implemented in SciPy (*Virtanen et al., 2020*). The complete linkage criterion also gave reasonable clustering. This agglomerative metric assigns clusters by successively combining clusters that minimize the sum of squared errors between them. Hierarchical clustering presents an advantage here because it does not assume the number of clusters a priori. Rather, final clusters were selected using the Ward distances showed in the dendrograms (see supplemental) as a guide and by overlaying the ligand occupancy density on the protein to ensure that each cluster represents a distinct spatial region of the protein.

## Quantifying residue similarity with the overlap coefficient (Szymkiewicz–Simpson coefficient)

To quantify the similarity between two sets of residues $A$ and $B$, the overlap coefficient was computed by dividing the number of overlapping residues between $A$ and $B$ by the size of the smaller set of residues and is illustrated in the equation below (*Vijaymeena and Kavitha, 2016*):

$$OC(A,B) = \frac{|A \cap B|}{min(|A|,|B|)}$$

Scaling the size of the intersection by the smallest set size normalizes the overlap and accounts for the large range in pathway lengths. If $A$ is a subset of $B$, then $OC(A,B) = 1$. This scaling method is appropriate, since these pathways are stochastic and involve a mixture of random residue contacts and 'guiding' residue contacts critical for binding. This would be problematic for the more common Jaccard similarity metric, which scales the intersection by the total size of both sets, where many random contacts increase pathway length and dilute the value of the similarity metric.

The overlap coefficient was used to quantify the residue overlap between pairs of pathways in each cluster to validate the spatial clustering and determine whether pathways within clusters involve similar residue contacts. In addition, this metric was used to quantify the similarity between residues involved in D-serine and glutamate binding.

## Umbrella sampling

All-atom models were constructed from monomeric GluN1 (PDB ID: 1PB8 *Furukawa and Gouaux, 2003*) and GluN2A (based on PDB ID: 2A5S *Furukawa et al., 2005*). Since no crystal structure of D-serine bound GluN2A exists, LBDs were constructed using MODELLER (*Webb and Sali, 2016*) to fill in missing residues, and sidechain remodeling was performed on those residues using SCWRL4 (*Krivov et al., 2009*). D-serine and glycine were modeled into the GluN2A LBD by superimposing the conserved arginine of the 2A5S glutamate-bound crystal structure (Arg-518) with the conserved arginine of the D-serine (1PB8) or glycine (1PB7) bound crystal structure, since there exists no crystal structure of GluN2A bound to these agonists. Bound crystallographic waters in the GluN2A (2A5S) and GluN1 (1PB8) structures were retained in the simulations.

To generate windows for umbrella sampling, targeted molecular dynamics simulations were performed by 'opening' the closed LBD along the order parameter $(\xi_1, \xi_2)$ (*Yao et al., 2013*). Specifically, $\xi_1$ and $\xi_2$ are defined as the center of mass distance between the backbone atoms of the following residue selections: $\xi_1$ is defined by residues 484–485 and 688–689 for GluN1 and residues 485–486 and 689–690 for GluN2A. $\xi_2$ is defined by residues 405–407 and 714–715 for GluN1 and 413–414 and 713–714 for GluN2A. 205 simulation windows were selected at 1 Å ×1 Å increments. Each window was solvated with a solvent box with dimensions 94 Å ×72 Å ×68 Å and 150 mM NaCl.

Umbrella sampling simulations were performed by applying a bias of 2 kcal mol$^{-1}$ Å$^{-2}$ to the $(\xi_1, \xi_2)$ order parameter to each of the 205 simulation windows. Equilibration was performed in an NVT ensemble by gradually relaxing backbone and sidechain restraints, and production simulations were carried out in an NPT ensemble at 300 K and 1 atm for best comparison with previously computed NMDAR LBD monomers (*Yao et al., 2013*). To ensure that the agonist does not diffuse out of the binding site, a restraint of 2 kcal mol$^{-1}$ Å$^{-2}$ between the carboxyl group of the agonist and the guanidinium group of the conserved arginine (Arg-523 for GluN1 and Arg-518 for GluN2A) was applied if the distance between these groups exceeded 3.2 Å. Previous work has indicated that these restraints do not affect the results but ensures that only the bound population is sampled (*Yao et al., 2013*). A weak center-of-mass restraint of 0.5 kcal mol$^{-1}$ Å$^{-2}$ was used applied to the N, CA, and C atoms of residues 461–463, 507–509, and 523–525 for GluN2A and residues 460–462, 512–514, and 528–530 for GluN1 to prevent translational protein motion. Biased trajectories were mathematically unbiased using the weighted histogram analysis method (WHAM) (*Kumar et al., 1992*; *Souaille et al., 2001*). 5 ns of production sampling for each window were used to compute the potential of mean force (PMF) for each simulation agonist. Standard deviations of all PMFs were computed by block averaging with ten blocks of trajectory for each window (*Grossfield and Zuckerman, 2009*).

## Computing energetics of glycan conformational dynamics

To quantify glycan conformational dynamics, a glycan-D2 order parameter was defined as the minimum distance between the heavy atoms of the glycans near the binding cleft (N491-Man5 for GluN1 and N443-Man5 and N444-Man5 for GluN2A) and the bottom lobe $C_\alpha$ atoms (residues 537–544 and 663–754 for GluN1 and residues 533–539 and 661–757 for GluN2A). One relative PMF was computed for each of the three near-pocket glycans using a window size of 0.2 Å using all glycosylated datasets. Error for each PMF was quantified using the standard deviation computed by block averaging with five blocks (*Figure 3—figure supplement 2A-D*). Blocks for which the window is not sampled were omitted from the error calculation; this was only necessary for high glycan distances >20 Å. A 1D projection of the $(\xi_1, \xi_2)$ order parameter, $\xi_{12}$, which averages $\xi_1$ and $\xi_2$, was used as a single measure of LBD closure for computing glycan PMFs (*Yao et al., 2013*; *Wied et al., 2019*; *Chin et al., 2020*).

## Electrophysiology

cRNA encoding GluN1-4a and GluN2A was injected into defolliculated *Xenopus laevis* oocytes (0.2–0.5 ng total cRNA per oocyte). The oocytes were incubated in recovery medium (50% L-15 medium (Hyclone) buffered by 15 mM Na-HEPES at a final pH of 7.4), supplemented with 100 µg mL$^{-1}$ streptomycin, and 100 U mL$^{-1}$ penicillin at 18 °C. Two electrode voltage clamp (TEVC; Axoclamp-2B) recording was performed between 24 and 48 hr after injection using an extracellular solution containing 5 mM HEPES, 100 mM NaCl, 0.3 mM BaCl$_2$, 10 mM Tricine at final pH 7.4 (adjusted with KOH). The current was measured using agarose-tipped microelectrode (0.4–0.9 MΩ) at the holding potential of −60 mV. Maximal response currents were evoked by 100 µM of D-serine and 100 µM of L-glutamate. Data was acquired by the program PatchMaster (HEKA) and analyzed by Origin 8 (OriginLab Corp).

## Acknowledgements

Anton 2 computer time (MCB130045P) was provided by the Pittsburgh Supercomputing Center (PSC) through NIH grant R01GM116961 (to AYL); the Anton 2 machine at PSC was generously made available by DE Shaw Research. We also used resources provided by the Maryland Advanced Research Computing Center (MARCC) at Johns Hopkins University. This work was funded by the Johns Hopkins Catalyst Award (to AYL); NIH T32GM135131 (to RAY and SJB); NIH NS111745 and MH085926 (to HF); Robertson funds at CSHL, Doug Fox Alzheimer's fund, Austin's purpose, Heartfelt Wing Alzheimer's fund, and the Gertrude and Louis Feil Family Trust (to HF).

## Additional information

### Funding

| Funder | Grant reference number | Author |
|---|---|---|
| National Institutes of Health | T32GM135131 | Remy A Yovanno Sarah J Brantley |
| National Institutes of Health | NS111745 | Hiro Furukawa |
| National Institutes of Health | MH085926 | Hiro Furukawa |
| Robertson Funds at CSHL | | Hiro Furukawa |
| Doug Fox Alzheimer's Fund | | Hiro Furukawa |
| Austin's Purpose | | Hiro Furukawa |
| Heartfelt Wing Alzheimer's Fund | | Hiro Furukawa |
| Gertrude and Louis Feil Family Trust | | Hiro Furukawa |
| Johns Hopkins Catalyst Award | | Albert Y Lau |

The funders had no role in study design, data collection and interpretation, or the decision to submit the work for publication.

### Author contributions

Remy A Yovanno, Conceptualization, Data curation, Formal analysis, Investigation, Visualization, Methodology, Writing - original draft, Writing - review and editing; Tsung Han Chou, Data curation, Formal analysis, Investigation, Visualization, Methodology, Writing - original draft, Writing - review and editing; Sarah J Brantley, Formal analysis; Hiro Furukawa, Resources, Formal analysis, Supervision, Funding acquisition, Investigation, Methodology, Writing - original draft, Project administration, Writing - review and editing; Albert Y Lau, Conceptualization, Resources, Formal analysis, Supervision, Funding acquisition, Investigation, Methodology, Writing - original draft, Project administration, Writing - review and editing

### Author ORCIDs

Remy A Yovanno http://orcid.org/0000-0003-3852-6684
Tsung Han Chou http://orcid.org/0000-0001-6154-6283
Hiro Furukawa http://orcid.org/0000-0001-8296-8426
Albert Y Lau http://orcid.org/0000-0002-0967-7558

### Decision letter and Author response

Decision letter https://doi.org/10.7554/eLife.77645.sa1
Author response https://doi.org/10.7554/eLife.77645.sa2

## Additional files

### Supplementary files

• Transparent reporting form

### Data availability

Data Availability: Data for Figs. 1, 2, and 5 (residue contact frequencies) are included with the manuscript in the source data. Code to perform pathway similarity analysis, to analyze binding pathways for D-serine and glutamate, and to compute glycan PMFs was uploaded to the Dryad link below (Figs. 1, 2, 5, 6, and 7). Also included at the Dryad link are the PMF data (Figs. 3, 6, and 7), electrophysiology data (Fig. 4), and MD trajectories.

The following dataset was generated:

| Author(s) | Year | Dataset title | Dataset URL | Database and Identifier |
|---|---|---|---|---|
| Yovanno RA, Chou T, Brantley S, Furukawa H, Lau A | 2022 | Data for: Excitatory and inhibitory D-serine binding to the NMDA receptor | https://dx.doi.org/10.5061/dryad.ns1rn8pwz | Dryad Digital Repository, 10.5061/dryad.ns1rn8pwz |

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
