## [Editor Report]

Activation of NMDA receptors requires two co-agonists: Glutamate that binds to the GluN2 subunit and glycine/D-serine that binds to the GluN1 subunit. In the present manuscript, the authors address the interaction of D-serine, which is a less studied co-agonist than glycine, with the GluN1 and GluN2A subunits using molecular simulations as well as electrophysiology experiments. Surprisingly they find that D-serine interacts with the GluN2 subunit, further expanding our molecular understanding of NMDA receptor structure-function. This paper will be of interest to those who study NMDA receptors and ligand-gated ion channels in general.

---

## [Decision Letter]

**Decision letter after peer review:**

Thank you for submitting your article "Excitatory and inhibitory D-serine binding to the NMDA receptor" for consideration by *eLife*. Your article has been reviewed by 3 peer reviewers, including Janice L Robertson as Reviewing Editor and Reviewer #1, and the evaluation has been overseen by josé Faraldo-Gómez as the Senior Editor. The following individual involved in the review of your submission has agreed to reveal their identity: Yun Lyna Luo (Reviewer #2).

Essential revisions:

The reviewers thought the manuscript was interesting and the research was carefully conducted. In particular, the discovery of D-serine interaction with GluN2A and its inhibitory effect is a novel result that will be of interest to the broader community. However, some questions remained as to the mechanism and the physiological or pharmacological relevance. The following major revisions are required to clarify these questions.

1) It is unclear to us whether D-serine has the capacity to reach such high concentrations in a physiological or pharmacological setting. Please provide more justification for this, or attenuate the conclusions that this provides a possible therapeutic treatment.

2) Extensive analysis is presented about the association of D-serine and its impact on LBD closure or efficacy. However, differences in agonist potency can be due to the differences in binding affinity and/or efficacy. Stabilization of the closed LBD conformation may indicate a change in efficacy, but affinity (KD) will still play a role in the final potency. The question still remains as to whether the binding affinity of D-serine to the two LBDs is stronger or weaker in comparison with glutamate and glycine. The relative strength of binding may be estimated if multiple associations and dissociation events have been captured in the conventional MD simulations. But it is not really clear whether dissociation events have been observed, and this needs to be clearly presented in the revised manuscript. Alternatively, this can be computed using alchemical free energy calculations or PMF calculations. Finally, an experimental KD should be extracted from the experimental competition data to compare to glutamate binding affinity and provide a reference for the computational analysis.

3) It is proposed that guided-diffusion drives serine binding to its site. This would imply that the residues on this path are necessary, and if mutated, would decrease the association rate and the ability for D-serine to compete with glutamate. Additional electrophysiological experiments or direct binding experiments would be useful in understanding the relevance of guided diffusion in the ligand-binding mechanism of NMDARs.

4) Please clarify what is the non-specific association signal in the MD simulations. Perhaps this has already been addressed in a previous study but should be included here. One option is to analyze the current trajectories and calculate the association event probabilities for a residue on the proposed guided path, compared to a similar residue at another interface that does not lead to the binding site. Alternatively, one could compare the current results with a negative control simulation where the ligand was replaced with a similar amino acid or molecule that has been verified as a non-binder for NMDAR.

*Reviewer #1 (Recommendations for the authors):*

The following changes are suggested for clarification:

1. The supplementary figure labels do not match the text.

*Reviewer #2 (Recommendations for the authors):*

The 2D-PMF of apo-state GluN2A LBD (Figure 3 C) only shows one minimum, rather than two states (open vs. closed) separated by a free energy barrier. Some clarification would be helpful for readers to better understand this free energy landscape.

On page 9 line 186, "we identified residues critical for stabilizing the agonist in the closed state by analyzing contacts in lowest-energy (<1 kcal/mol) conformers". More information is needed here or in the method section in terms of how the lowest energy was computed.

On page 10 line 217, "In fact, we observed D-serine binding to GluN2A, even in presence of glutamate" in the bulk solution or in the binding site? This is an important point if the LBD could accommodate glutamate and D-serine at the same time. But this somehow contradicts the competitive binding mechanism. If glutamate is present in the bulk solution during D-serine spontaneous binding simulations, do they have the same bulk concentration? Please clarify.

On page 10 line 218, "glutamate bound more frequently than D-serine and with longer residence times in the binding site" While the raw data is available in the Datasets, the number of binding events and residence time (1/Koff) could be briefly mentioned here to give a more quantitative comparison.

On page 17 lines 381-382, Figure S7 should be Figure S6?

*Reviewer #3 (Recommendations for the authors):*

I have just some general comments:

1. Just a suggestion but for Figures 1 and 2, it might be nice to have each figure panel indicate what it is showing. For example, above Figure 1C one could have a header 'Xi2 face dominates'. Figure 1D: 'Xi1 face of D1 lobe dominates'. Etc. Right now one must look back and forth between the legend/Results section and figure to discern what specifically is being shown. Again this is just a suggestion.

2. Electrophysiological experiments. The effect of D-serine, as noted by the authors, only occurs at fairly high concentrations especially relative to glutamate. The authors conclude that this reflects competition between glutamate and D-serine for GluN2 binding site. Might D-serine or glycine have alternative effects on receptor function? For example, do not these ligands induce receptor desensitization?

3. Given the fairly high concentrations of D-serine especially relative to glutamate, I am not certain that there would be any physiological or even pharmacological (i.e., D-serine as a drug treatment) impact. Either justify these comments more or attenuate them.

4. The N-glycans simulations are interesting and further expand our molecular insight into agonist/binding site interactions. However, many of the results are shown in Supplemental Material. Also, it would be helpful to have a summary figure of these results. Right now the information is buried in the text and it is hard to discern the conclusion of these experiments without reading and rereading to identify the outcome.

Related question. Figure 6C-6E. I see how GluN2A N443-Man5 and GluN1 N491-Man5 show an increased PMF more proximal to the D2 lobe. However, do these interactions impact D1:D2 interactions? Cannot this be assayed using the two-dimensional order parameter (Xi1:Xi2)?

---

## [Author Response]

Essential revisions:The reviewers thought the manuscript was interesting and the research was carefully conducted. In particular, the discovery of D-serine interaction with GluN2A and its inhibitory effect is a novel result that will be of interest to the broader community. However, some questions remained as to the mechanism and the physiological or pharmacological relevance. The following major revisions are required to clarify these questions.(1) It is unclear to us whether D-serine has the capacity to reach such high concentrations in a physiological or pharmacological setting. Please provide more justification for this, or attenuate the conclusions that this provides a possible therapeutic treatment.

We have attenuated the conclusions by removing the sentence, “If true, this behavior may factor into therapeutic strategies focused on increasing D-serine concentration in the synapse by establishing an upper dosage limit after which a D-serine increase is no longer potentiating” on page 11. The relationship between dosage and synaptic concentration is complicated and depends upon a variety of factors including the delivery method and uptake efficiency. However, since D-serine is being pursued as a supplemental treatment for neurological disorders, the finding that D-serine competes with glutamate for binding to GluN2A is still important to note.

(2) Extensive analysis is presented about the association of D-serine and its impact on LBD closure or efficacy. However, differences in agonist potency can be due to the differences in binding affinity and/or efficacy. Stabilization of the closed LBD conformation may indicate a change in efficacy, but affinity (KD) will still play a role in the final potency. The question still remains as to whether the binding affinity of D-serine to the two LBDs is stronger or weaker in comparison with glutamate and glycine. The relative strength of binding may be estimated if multiple associations and dissociation events have been captured in the conventional MD simulations. But it is not really clear whether dissociation events have been observed, and this needs to be clearly presented in the revised manuscript. Alternatively, this can be computed using alchemical free energy calculations or PMF calculations. Finally, an experimental KD should be extracted from the experimental competition data to compare to glutamate binding affinity and provide a reference for the computational analysis.

The supplemental datasets provided with the manuscript specify the exact number of association (B) events and dissociation (U) events. We added a line on page 5 of the main text to clarify the presence of both event types. We used the following expression (Pan et al., *J. Chem. Theory Comput.,* 2017) for computing direct binding free energy ΔG_b_ from K_D_: KD=PuPbvcoNA and ΔG_b_ = −k_B_T ln K_D_ While this approach does produce a value, the values obtained do not provide a reasonable estimate of binding affinity. This is due to the absence of full LBD closure following each association event, resulting in an undersampling of the fully closed LBD. The amount of time the agonist spends in the binding site is partially determined by the degree of LBD closure. As a result, the ratio of unbound and bound frames varies greatly across all simulations, making the direct calculation of ΔG_b_ from our equilibrium MD not feasible. The same issue affects the calculation of an accurate k_off_. For this reason, we performed umbrella sampling simulations to separately assess the thermodynamics of LBD closure in the presence of different agonists, giving relative free energy differences between LBD conformations. Experimentally, all work in this manuscript is done with full-length receptors, so EC50/IC50 is what can be measured.

(3) It is proposed that guided-diffusion drives serine binding to its site. This would imply that the residues on this path are necessary, and if mutated, would decrease the association rate and the ability for D-serine to compete with glutamate. Additional electrophysiological experiments or direct binding experiments would be useful in understanding the relevance of guided diffusion in the ligand-binding mechanism of NMDARs.

To address this point, we performed additional TEVC experiments generating D-serine dose-response curves for GluN1a Arg694Ala and Arg695Ala, and GluN2A Arg692Ala and Arg695Ala. The curves for both GluN2A mutants support our guided diffusion mechanism, as they lowered the D-serine inhibition potency (These mutants also likely also alter glutamate binding, but since D-serine and glutamate bind through the same residues, it is not possible to separate out individual contributions.) The GluN1a mutants did not show altered behavior, supporting the increased diffusiveness of D-serine binding to GluN1 compared to GluN2A. These additional findings are included in the main text on page 12 and in Figure 4D.

(4) Please clarify what is the non-specific association signal in the MD simulations. Perhaps this has already been addressed in a previous study but should be included here. One option is to analyze the current trajectories and calculate the association event probabilities for a residue on the proposed guided path, compared to a similar residue at another interface that does not lead to the binding site. Alternatively, one could compare the current results with a negative control simulation where the ligand was replaced with a similar amino acid or molecule that has been verified as a non-binder for NMDAR.

Calculating the number of successful binding events compared with random associations allows us to determine the level of noise present in the binding process. For glycosylated simulations with 19.6 mM D-serine, we observe an average of 1242 ± 31 random GluN2A associations (n=3 simulations) per microsecond (1249 ± 1 for GluN1, n=2 simulations) that fail to result in successful binding. In these same simulations, we observe about 1.2 ± 0.6 successful GluN2A binding events per microsecond (1.0 ± 0.5 for GluN1).

Supporting our guided-diffusion mechanism, we identified residues for which the ratio of successful to random binding was increased. In general, GluN2A LBD residues contacted by the agonist experience 26 ± 2 random associations per microsecond (31 ± 1 for GluN1). For each residue involved in successful D-serine binding pathways, we calculated the percentage of associations resulting in successful binding. For residues important for guided-diffusion pathways, this percentage is >1% (a column was added to Figure 1–source data 4 and Figure 2–source data 3). This allows us to quantify the importance of pathway residues despite a noisy non-specific association signal.

This analysis has been added to page 16 of the main text.

Reviewer #1 (Recommendations for the authors):The following changes are suggested for clarification:1. The supplementary figure labels do not match the text.

Thank you. We have made the corrections.

Reviewer #2 (Recommendations for the authors):The 2D-PMF of apo-state GluN2A LBD (Figure 3 C) only shows one minimum, rather than two states (open vs. closed) separated by a free energy barrier. Some clarification would be helpful for readers to better understand this free energy landscape.

Thank you for the suggestion. A line guiding the interpretation of this PMF was added to the main text on page 9.

On page 9 line 186, "we identified residues critical for stabilizing the agonist in the closed state by analyzing contacts in lowest-energy (<1 kcal/mol) conformers". More information is needed here or in the method section in terms of how the lowest energy was computed.

The lowest-energy conformers were extracted from the 2D PMF computed using umbrella sampling simulations. A line clarifying this was added to the main text on page 9.

On page 10 line 217, "In fact, we observed D-serine binding to GluN2A, even in presence of glutamate" in the bulk solution or in the binding site? This is an important point if the LBD could accommodate glutamate and D-serine at the same time. But this somehow contradicts the competitive binding mechanism. If glutamate is present in the bulk solution during D-serine spontaneous binding simulations, do they have the same bulk concentration? Please clarify.

Glutamate and D-serine cannot bind the GluN2A LBD at the same time, as the binding is competitive. This is clarified on page 4 to avoid confusion.

On page 10 line 218, "glutamate bound more frequently than D-serine and with longer residence times in the binding site" While the raw data is available in the Datasets, the number of binding events and residence time (1/Koff) could be briefly mentioned here to give a more quantitative comparison.

For our 15 μs glycosylated simulation with both D-serine and glutamate agonists (9.8 mM each agonist), we observed 17 glutamate binding events with an average time bound of 131 ns. In the same simulation, we observed 5 D-serine binding events with an average time bound of 3 ns. For our 15 μs non-glycosylated simulation with both D-serine and glutamate agonists (9.8 mM each agonist), we observed 75 glutamate binding events with an average time bound of 236 ns and 6 D-serine binding events with an average time bound of 56 ns. This information was added to the main text on page 11.

On page 17 lines 381-382, Figure S7 should be Figure S6?

Thank you. Figure labels were corrected in the main text.

Reviewer #3 (Recommendations for the authors):I have just some general comments:1. Just a suggestion but for Figures 1 and 2, it might be nice to have each figure panel indicate what it is showing. For example, above Figure 1C one could have a header 'Xi2 face dominates'. Figure 1D: 'Xi1 face of D1 lobe dominates'. Etc. Right now one must look back and forth between the legend/Results section and figure to discern what specifically is being shown. Again this is just a suggestion.

Figures 1 and 2 were revised accordingly.

2. Electrophysiological experiments. The effect of D-serine, as noted by the authors, only occurs at fairly high concentrations especially relative to glutamate. The authors conclude that this reflects competition between glutamate and D-serine for GluN2 binding site. Might D-serine or glycine have alternative effects on receptor function? For example, do not these ligands induce receptor desensitization?

It is known that D-serine or glycine alone does not activate the NMDAR channel activity. Both glutamate and D-serine or glycine are required for activity. In general, a low concentration of glycine (or D-serine) has a negative effect as increased glycine (or D-serine) concentration can mask the effect. Thus, a high concentration of D-serine most likely does not induce desensitization. Furthermore, the MD simulations and electrophysiology do not support allosteric inhibition.

3. Given the fairly high concentrations of D-serine especially relative to glutamate, I am not certain that there would be any physiological or even pharmacological (i.e., D-serine as a drug treatment) impact. Either justify these comments more or attenuate them.

Please see the response to Essential Revisions #1 above.

There have been a number of reports of activity-dependent D-serine release from both neurons and glia (e.g., Rosenberg et al., FASEB J, 2010). They involved transporters and channels such as VRAC. While the local concentration of D-serine at synapses or extrasynaptic space has not been measured precisely, the active D-serine transport could raise its concentration. While full channel inhibition by D-serine is unlikely, competitive D-serine inhibition may occur partially in the physiological environment.

4. The N-glycans simulations are interesting and further expand our molecular insight into agonist/binding site interactions. However, many of the results are shown in Supplemental Material. Also, it would be helpful to have a summary figure of these results. Right now the information is buried in the text and it is hard to discern the conclusion of these experiments without reading and rereading to identify the outcome.Related question. Figure 6C-6E. I see how GluN2A N443-Man5 and GluN1 N491-Man5 show an increased PMF more proximal to the D2 lobe. However, do these interactions impact D1:D2 interactions? Cannot this be assayed using the two-dimensional order parameter (Xi1:Xi2)?

In Figure 6 —figure supplement 2, we calculate the glycan-D2 distance PMF as a function of a one-dimensional projection of the Xi1,Xi2 order parameter and show that the closed LBD results in a steeper glycan-D2 PMF. This suggests that glycan-D2 interactions favor D1-D2 interactions. To emphasize this as the main finding regarding glycans, we moved this to the main figures as Figure 7.